# Auxin apical dominance governed by the OsAsp1-OsTIF1 complex determines distinctive rice caryopses development on different branches

**Shu Chang[1], Yixing Chen[1], Shenghua Jia[1], Yihao Li[1], Kun Liu[1], Zhouhua Lin[1], Hanmeng Wang[1], Zhilin Chu[1], Jin Liu[1], Chao Xi[1], Heping Zhao[1], Shengcheng Han[1,2]\*, Yingdian Wang[1,2]\***

**1** Beijing Key Laboratory of Gene Resource and Molecular Development, College of Life Sciences, Beijing Normal University, China, **2** Academy of Plateau Science and Sustainability of the People's Government of Qinghai Province & Beijing Normal University, Qinghai Normal University, Qinghai, China

\* schan@bnu.edu.cn (SH); ydwang@bnu.edu.cn (YW)

**Data Availability Statement:** Sequence data in this article can be found in the http://rice.plantbiology.msu.edu/index.shtml database under the following accession codes: OsAsp1 (Os11g08400), OsCycB1

## Abstract

In rice (*Oryza sativa*), caryopses located on proximal secondary branches (CSBs) have smaller grain size and poorer grain filling than those located on apical primary branches (CPBs), greatly limiting grain yield. However, the molecular mechanism responsible for developmental differences between CPBs and CSBs remains elusive. In this transcriptome-wide expression study, we identified the gene *Aspartic Protease 1* (*OsAsp1*), which reaches an earlier and higher transcriptional peak in CPBs than in CSBs after pollination. Disruption of *OsAsp1* expression in the heterozygous T-DNA line *asp1-1*[+/−] eliminated developmental differences between CPBs and CSBs. OsAsp1 negatively regulated the transcriptional inhibitor of auxin biosynthesis, *OsTAA1* transcriptional inhibition factor 1 (OsTIF1), to pre-serve indole-3-acetic acid (IAA) apical dominance in CPBs and CSBs. IAA also facilitated OsTIF1 translocation from the endoplasmic reticulum (ER) to the nucleus by releasing the interaction of OsTIF1 with OsAsp1 to regulate caryopses IAA levels via a feedback loop. IAA promoted transcription of *OsAsp1* through MADS29 to maintain an OsAsp1 differential between CPBs and CSBs during pollination. Together, these findings provide a mechanistic explanation for the distributed auxin differential between CPBs and CSBs to regulate distinct caryopses development in different rice branches and potential targets for engineering yield improvement in crops.

## Author summary

Rice is a major food crop and an important model plant. Compared with caryopses on apical primary branches (CPBs) of rice, those located on proximal secondary branches (CSBs) display smaller grains and poor grain filling, which greatly limit rice yield potential fulfilment, especially among 'super' rice cultivars. In this study, we demonstrated that high indole-3-acetic (IAA) levels upregulated *Aspartic Protease 1* (*OsAsp1*) transcription

(Os01g599120), OsKRP1 (Os02g52480), OsCCS52A (Os03g03150), OsTAA1 (Os01g07500), OsYUCCA1 (Os01g45760), OsTIF1 (Os04g02510), OsTIF2 (Os04g59380), MADS29 (Os02g07340) and OsActin1 (Os03g50885). The transcriptome data have been deposited at the Sequence Read Archive under accession number SRP083496.

**Funding:** This work was supported by the National Key Basic Research Program of China (973 Program, 2013CB126902) and the National Natural Science Foundation of China (grant nos. 30570148 and 31370307). The funders had no role in study design, data collection and analysis, decision to publish, or preparation of the manuscript.

**Competing interests:** The authors have declared that no competing interests exist.

via MADS29 post-pollination to produce higher *OsAsp1* levels in CPBs than in CSBs. OsAsp1 then interacted with *OsTAA1* transcriptional inhibition factor 1 (OsTIF1) in the endoplasmic reticulum (ER) to dispel OsTIF1 transcriptional inhibition of *OsTAA1*, causing IAA content to peak in CPBs at 5 days after fertilisation (DAF). IAA facilitated OsTIF1 translocation from the ER to the nucleus by reducing its interaction with OsAsp1 as feedback regulation of IAA levels in caryopses. Thus, differential auxin levels between CPBs and CSBs are determined by the OsAsp1-OsTIF1 complex, and are essential for the distinct development of CPBs and CSBs, providing potential targets for engineering yield improvement in crops.

## Introduction

Rice (*Oryza sativa*) is a staple food for approximately two-thirds of the global population. Due to the application of dwarf breeding and hybrid breeding in the past 50 years, rice grain yield has more than doubled in most parts of the world, and even tripled in certain countries and regions [1]. However, these newer rice cultivars have not entirely fulfilled their high-yield potential due to smaller grain size and poorer grain filling in caryopses located on proximal secondary branches (CSBs) than in those of apical primary branches (CPBs). This problem is exacerbated in recently bred 'super' rice cultivars [2].

Rice caryopsis development begins with a double-fertilisation event, following which endosperm mainly contributes to grain size and weight in two continuous developmental progresses: early-stage endosperm cell proliferation and differentiation, and late-stage grain filling. Extensive studies have shown that grain size and weight on a single spikelet are regulated by the histone acetylation and deacetylation pathway [3], the ubiquitin–proteasome pathway [4–6], G-protein signalling [7, 8], MAPK signalling [9], and phytohormone signalling, such as that involving brassinosteroids [10–12] or auxins [13, 14], which influence spikelet hull and endosperm growth. Grain filling is associated with carbohydrate supply, activity of the enzymes involved in starch metabolism, levels of various endogenous hormones, and environmental conditions [15–18]. Studies using omics approaches have revealed differences in the expression of various proteins [19], in the expression of genes related to starch synthesis and hormone signalling [20], and in miRNA levels between CPBs and CSBs during the middle and late stages of grain filling after fertilisation [21, 22]. However, the molecular mechanisms determining the developmental differences between CPBs and CSBs remain elusive. In this transcriptome-wide expression study, we sought to identify the key genes that are differentially expressed in post-anthesis CPBs and CSBs, and further characterised the role of one of these genes, *Aspartic Protease 1* (*OsAsp1*), in determining the distinctive development of CPBs and CSBs in rice.

## Results

### Transcriptome-wide expression analysis of CPBs and CSBs to identify *OsAsp1*

To identify the key factors that determine the developmental differences between CPBs and CSBs in rice, we performed a transcriptome-wide expression analysis of CPB and CSB samples obtained at four different dates. Samples were chosen separately for each branch type to ensure morphological similarity of CPBs and CSBs, given that primary branches develop earlier than secondary branches. Specifically, samples were collected at 0, 5, 12, and 20 days after heading

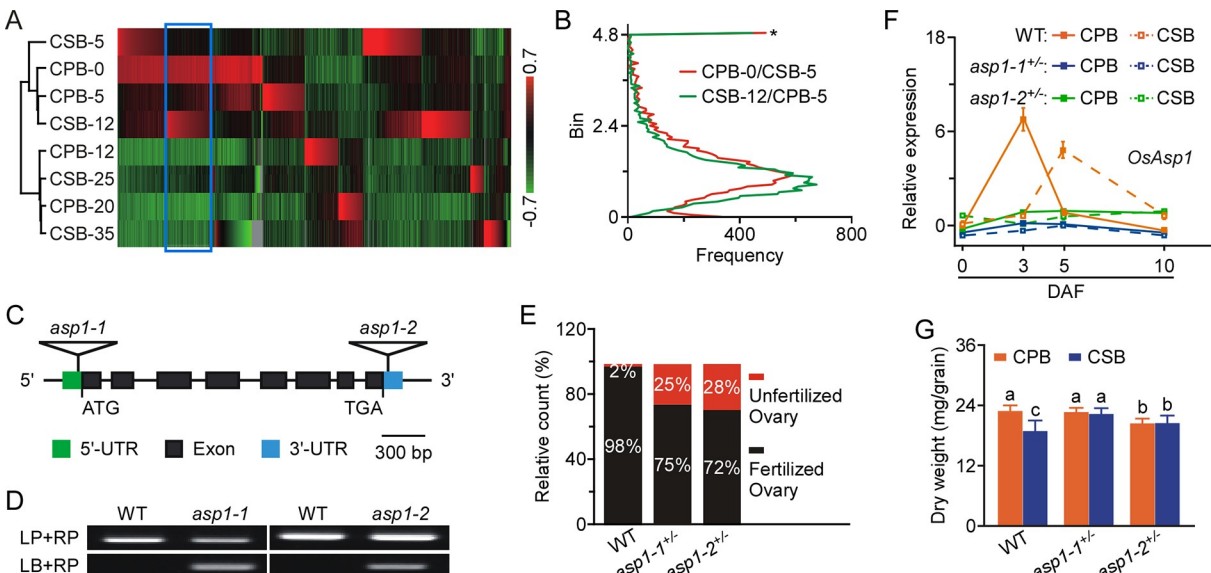

**Fig 1. Disruption of differential *OsAsp1* expression eliminated the distinct development of caryopses on different branches. A**, Heat map showing differentially expressed genes in CPBs at 0, 5, 12, and 20 days after heading (DAH) and in CSBs at 5, 12, 25, and 35 DAH. The gene expression profiles were deposited in the Sequence Read Archive (accession no. SRP083496). Blue rectangles on heat map indicate genes with higher transcriptional levels in CPB-0 and CSB-12 than in CSB-5 and CPBs5. Colour scale shows log ratio of fold change. **B**, Distributions of ratios of gene expression levels (blue rectangles, Fig 1A) based on the reads per kb per million mapped reads (RPKM) values for CPB-0 and CSB-5 (red line) and CSB-12 and CPB-5 (green line). *Ratio distribution of a significant coincidence for genes of CPB-0 vs. CSB-5 and CSB-12 vs. CPB-5, centred on a value of 4.8 ($\geq$ 4.8). **C**, Schematic representation of the two *OsAsp1* T-DNA insertions found in the *asp1-1* and *asp1-2* mutant rice lines. **D**, Identification of the genotypes of *asp1-1* and *asp1-2* by polymerase chain reaction (PCR). Wild-type (WT) rice was used as a control. LP, LB, and RP primers used for PCR are listed in S4 Table. **E**, Fertility of the WT, *asp1-1*$^{+/-}$, and *asp1-2*$^{+/-}$. Unfertilised and fertilised seeds were counted for each plant. **F**, Transcription levels of *OsAsp1* detected by quantitative reverse-transcription (qRT)-PCR of CPBs and CSBs in the WT, *asp1-1*, and *asp1-2* at 0, 3, 5, and 10 DAF. Solid and dotted lines indicate the transcript levels of *OsAsp1* in CPBs and CSBs, respectively. Red, blue, and green lines indicate the transcript levels of *OsAsp1* in the WT, *asp1-1*, and *asp1-2*, respectively. The expression of *OsAsp1* was normalised to that of *OsActin1*, and relative expression in CPBs in the WT at 0 DAF was set at 1.0. qRT-PCR data are means ± standard deviation (SD) of three independent experiments. **G**, Dry weights of developed grain from CPBs and CSBs in one whole plant of the WT, *asp1-1*$^{+/-}$, and *asp1-2*$^{+/-}$. Values are means ± SD (*n* = 30). Different letters indicate significant differences (*P* < 0.05, Student's *t*-test).

(DAH) for CPBs (CPB-0, -5, -12, and -20) and 5, 12, 25, and 35 DAH for CSBs (CSB-5, -12, -25 and -35) (S1A Fig).

The gene expression profiles formed two separate clusters, one consisting of CPB-0, CSB-5, CPB-5, and CSB-12 and the other of CPB-12, CSB-25, CPB-20, and CSB-35, among which two pairs, CPB-5/CSB-12 and CPB-20/CSB-35, were closer than the others (Fig 1A). This pattern is consistent with the existence of two continuous phases of caryopsis development: endosperm cell proliferation, which mainly contributes to grain enlargement, followed by grain filling. We also found that CPB endosperm had a higher cell reproduction rate and faster grain filling rate than that of CSBs (S1B and S1C Fig), which is consistent with the findings of previous studies [17, 23]. We propose that the difference in endosperm cell proliferation rates between CPBs and CSBs is the critical factor for grain production because it determines the sink capacity for caryopsis. Therefore, we further compared the genes that were more highly expressed in CPB-0 and CSB-12 than in CSB-5 and CPB-5 (Fig 1A) with the results of the gene expression ratio distribution analysis, based on reads per kb per million mapped reads (RPKM) values. Then, we identified candidate genes with expression ratios higher than 4.8 in both comparisons (Fig 1B and S1 Table) and used quantitative reverse-transcription polymerase chain reaction (qRT–PCR) to confirm the relative transcript levels of the top three genes (S2 Fig): *Os07g39020*, encoding the SUBTILISIN-LIKE PROTEASE OsSUB53; *Os06g41030*,

encoding an unknown protein containing a DUF1680 domain; and *OsAsp1*, a previously characterised nucellin gene (*aspartic protease1*, *Os11g08200*) [24]. Next, we characterised the role of *OsAsp1* in determining differential development between CPBs and CSBs.

## Disrupting *OsAsp1* eliminated CPB–CSB developmental differences

To explore the role of OsAsp1 in caryopsis development, we obtained two *OsAsp1* T-DNA mutation lines from the Rice Functional Genomic Express Database (RiceGE, http://signal. salk.edu): *asp1-1* (PFG_3A-60042) and *asp1-2* (PFG_3A-04889) (Fig 1C). Through self-crossing and back-crossing with wild-type (WT) plants, we confirmed that both *asp1-1* and *asp1*-2 are single-insertion and homozygous-lethal mutants, because about 75% of spikelets developed into fertilised seeds and none seeds of these seeds was homozygous (S2 Table, Fig 1D and 1E). We further detected *OsAsp1* transcript levels in CPBs and CSBs of the WT, *asp1-1*$^{+/−}$, and *asp1-2*$^{+/−}$ on four different dates, and found that *OsAsp1* expression peaked in CPBs at 3 days after flowering (DAF) and in CSBs at 5 DAF in the WT, and that the peak *OsAsp1* transcript level of WT CPBs at 3 DAF was higher than that of WT CSBs at 5 DAF (Fig 1F). However, in both CPBs and CSBs of *asp1-1*$^{+/−}$ and *asp1-2*$^{+/−}$, *OsAsp1* expression was stable and expression values at different DAF were similar to those at 0 DAF in both CPBs and CSBs of the WT (Fig 1F). These results indicate that T-DNA insertion decreased expression of *OsAsp1* in CPBs and CSBs in the independent lines *asp1-1*$^{+/−}$ and *asp1-2*$^{+/−}$. Then, we assessed the developmental parameters of both lines, and found that plant height was lower in both *asp1-1*$^{+/−}$ and *asp1-2*$^{+/−}$ than in the WT, but that spikelet number per panicle in *asp1-1*$^{+/−}$ was similar to that in the WT and greater than that in *asp1-2*$^{+/−}$ (S3 Fig). Of note, compared to the WT, the difference in the grain weight of the developed seeds between CPBs and CSBs disappeared in both *asp1-1*$^{+/−}$ and *asp1-2*$^{+/−}$, whereas seed grain weights in *asp1-1*$^{+/−}$ and *asp1-2*$^{+/−}$ were the same or closer to those of CPBs in the WT (S4 Fig and Fig 1G). These results indicate that OsAsp1 is essential for rice growth and caryopsis development; therefore, we focused on *asp1-1*$^{+/-}$ in the subsequent analysis.

Next, we observed the ovary development of CPBs and CSBs in the WT and *asp1-1*$^{+/−}$ at 3 and 5 DAF, respectively, and found that at both time points, *asp1-1*$^{+/−}$ displayed faster ovary development in CSBs than in the WT, and that the ovary size of CSBs in *asp1-1*$^{+/−}$ was similar to that of CPBs in both the WT and *asp1-1*$^{+/−}$ (Fig 2A and S5 Fig). We further used Eosin-B staining to assess the ovary development of CPBs and CSBs in the WT and *asp1-1*$^{+/−}$ at 3 and 5 DAF, and found that the CSBs in *asp1-1*$^{+/−}$ showed a higher level of endosperm cellularisation than either CSBs in the WT or CPBs in the WT and *asp1-1*$^{+/−}$ at both time points (Fig 2B). In addition, CSBs in *asp1-1*$^{+/−}$ displayed more developed embryos at 5 DAF than either CSBs in the WT or CPBs in the WT and *asp1-1*$^{+/−}$ (Fig 2B). Moreover, DNA ploidy analysis of CPBs and CSBs in the WT and *asp1-1*$^{+/−}$ at 3 and 5 DAF by flow cytometry showed that at both time points, the relative 3C plus 6C DNA contents, which represent the numbers of endosperm cells, were significantly higher in CSBs of *asp1-1*$^{+/−}$ than in CPBs of *asp1-1*$^{+/−}$ or either CPBs or CSBs of the WT, indicating that CSBs in *asp1-1*$^{+/−}$ developed better than did CPBs of *asp1-1*$^{+/−}$ or CPBs and CSBs of the WT (Fig 2C). Previous studies have identified *Orysa;CycB1;1* (*OsCycB1*), *Orysa;KRP1* (*OsKRP1*) and the rice cell cycle switch 52A gene (*OsCCS52A*) as three marker genes associated with cell cycle and division in rice endosperm [25–27]. qRT-PCR analysis showed that these three genes displayed differential expression patterns between CPBs and CSBs in the WT and *asp1-1*$^{+/−}$ at both 3 and 5 DAF (Fig 2D). Together, these data confirm that disruption of *OsAsp1* transcription accelerated CSBs' developmental progress, thereby relieving the developmental differences between CPBs and CSBs in *asp1-1*$^{+/−}$.

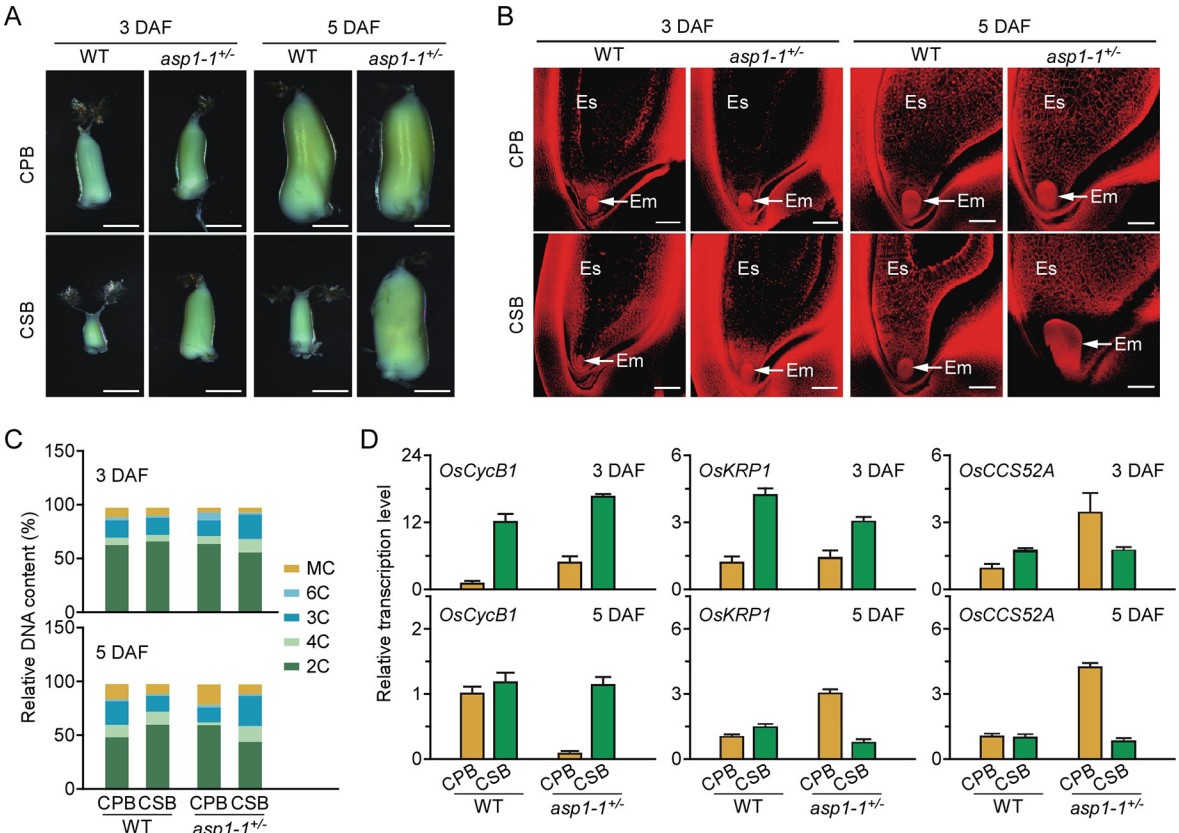

**Fig 2. OsAsp1 is required for differential ovary development in CPBs and CSBs. A**, Ovary phenotypes of CPBs and CSBs in the WT and *asp1-1*[+/−] at 3 and 5 days after flowering (DAF). Scale bars, 0.2 cm. **B**, Ovary developmental status of CPBs and CSBs in the WT and *asp1-1*[+/−] at 3 and 5 DAF detected by Eosin B staining. Em, embryo; Es, endosperm. Scale bars, 100 μm. **C**, Nuclear DNA ploidy distributions of cells from CPBs and CSBs (*n* = 6) of the WT and *asp1-1*[+/−] at 3 and 5 DAF detected by flow cytometry. MC indicates nuclear DNA ploidy exceeding 6C. **D**, Relative expression levels of *OsCycB1*, *OsKRP1*, and *OsCCS25A* in CPBs and CSBs of the WT and *asp1-1*[+/−] at 3 and 5 DAF detected by RT-qPCR. Target gene expression was normalised to that of *OsActin1*, and relative expression of the target gene in CPBs of the WT was set at 1.0. Values are means ± SD of three independent experiments.

## OsAsp1 activated *TAA1* transcription to upregulate IAA levels

A prior study showed that auxin biosynthesis couples central cell division for endosperm development to fertilisation in *Arabidopsis* [28]. Therefore, we monitored the levels of indole-3-acetic acid (IAA) in CPBs and CSBs of the WT and *asp1-1*[+/−] using high-performance liquid chromatography and found that in the WT, free IAA content was significantly higher in CPBs than in CSBs at 0, 3, and 5 DAF (Fig 3A), which is consistent with a previous report [16]. However, in *asp1-1*[+/−], there was no difference in IAA levels between CPBs and CSBs; IAA content was very low and almost identical in CPBs and CSBs at 0, 3, and 5 DAF (Fig 3A). In rice, IAA is mainly synthesised via the indole-3-pyruvic acid (IPA) pathway, catalysed by tryptophan aminotransferase (OsTAA) and flavin monooxygenase-like enzymes (OsYUCCAs) [29, 30]. As expected, in the WT, expression of *OsTAA1* gradually increased after fertilisation, reaching a peak at 5 DAF in both CPBs and CSBs, with a higher peak in CPBs. However, in *asp1-1*[+/−], the expression peaks of *OsTAA1* in both CPBs and CSBs disappeared at 5 DAF, but occurred at 3 DAF. Of note, *OsTAA1* expression was lower in CPBs than in CSBs of *asp1-1*[+/−] (Fig 3B). By contrast, the expression patterns of *OsYUCCA1*, which was more highly expressed in CPBs than in CSBs at 5 and 10 DAF, respectively, were similar in CPBs and CSBs in the WT and

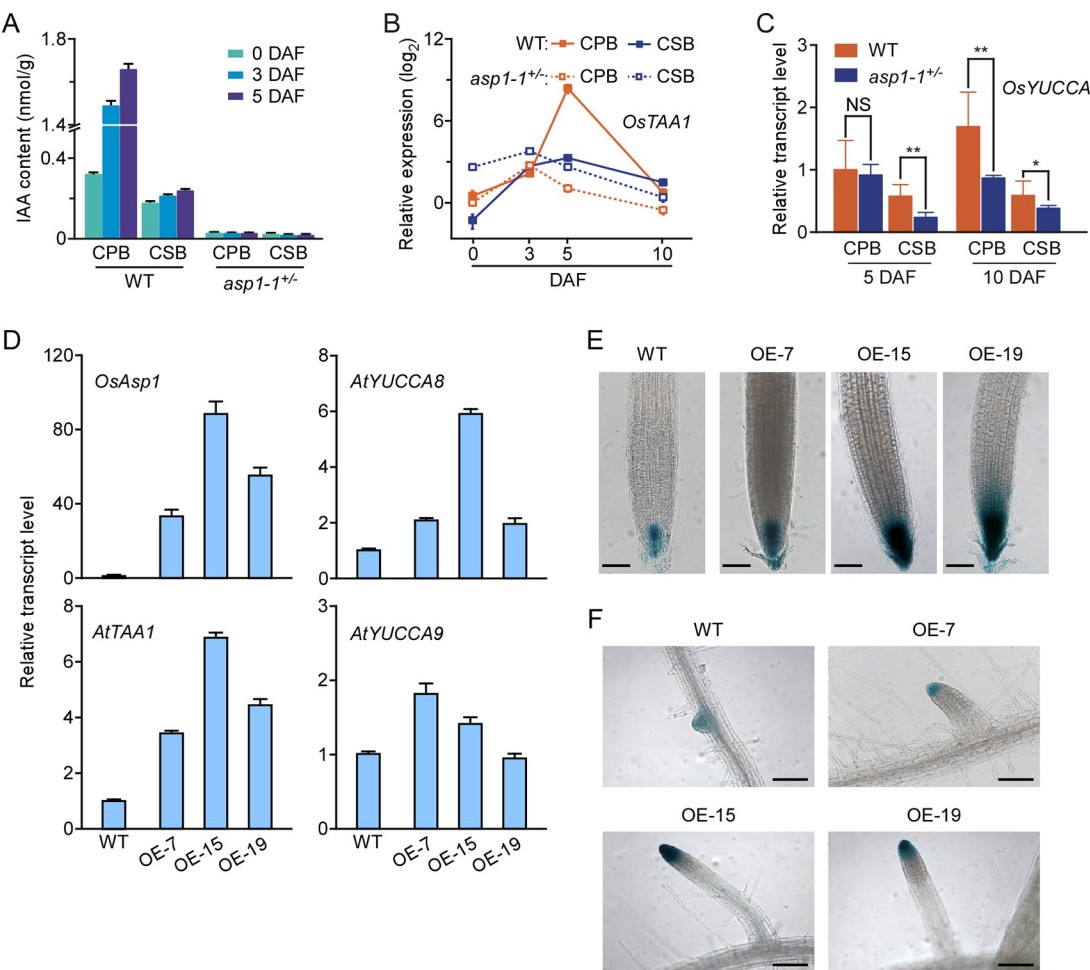

**Fig 3. OsAsp1 activates *TAA1* transcription to upregulate IAA levels. A**, IAA content of CPBs and CSBs of the WT and *asp1-1*$^{+/-}$ at 0, 3, and 5 DAF. Values are means ± SD of three independent experiments. **B**, Transcript levels of *OsTAA1* detected by qRT-PCR in CPBs and CSBs of the WT and *asp1-1*$^{+/-}$ at 0, 3, 5, and 10 DAF. *OsTAA1* expression was normalised to that of *OsActin1*, and relative expression in CPBs of the WT at 0 DAF was set at 1.0. qRT–PCR data are means ± SD of three independent experiments. **C**, Transcription levels of *OsYUCCA1* in CPBs and CSBs of the WT and *asp1-1*$^{+/-}$ on 5 and 10 DAF. *OsYUCCA1* expression was normalised to that of *OsActin1*, and relative expression of *OsYUCCA1* in CPBs of the WT at 5 DAF was set at 1.0. Values are means ± SD of three independent experiments. NS, no significant difference; $^*0.01 < P < 0.5$; $^{**}0.001 < P < 0.01$ (Student's t-test). **D**, Expression of *OsAsp1*, *AtTAA1*, *AtYUCCA8*, and *AtYUCCA9* in *Arabidopsis* WT and *OsAsp1-overexpressed* lines OE-5, OE-15, and OE-19. Target gene expression was normalised to that of *AtUBQ10*, and relative expression in the WT was set at 1.0. Values are means ± SD of three independent experiments. **E**, GUS staining of root tips from 3-day-old seedlings of *DR5: GUS Arabidopsis*, and *DR5:GUS* and *OsAsp1-overexpressed* crossed lines OE-5, OE-15, and OE-19. Scale bars, 20 µm. **F**, GUS staining of lateral roots of 3-day-old seedlings of *DR5:GUS*, and *DR5:GUS* and *OsAsp1-overexpressed* crossed lines OE-5, OE-15, and OE-19. Scale bars, 50 µm.

*asp1-1*$^{+/-}$, except that expression levels of *OsYUCCA1* were lower in CPBs of *asp1-1*$^{+/-}$ than in the WT at 10 DAF (Fig 3C). In addition, we heterologously expressed *OsAsp1* in *DR5::GUS* transgenic *Arabidopsis thaliana* plants, and found upregulated transcription of *AtTAA1* and *AtYUCCA8*, but not *AtYUCCA9* (Fig 3D); GUS activities were also strongly increased in root apical meristem of the main and lateral roots of 3-day-old seedlings (Fig 3E and 3F). These results imply that *OsTAA1* is upregulated by OsAsp1 to preserve IAA levels and ensure normal development of CPBs and CSBs.

## OsAsp1 dismissed transcriptional inhibition of OsTIF1 to *OsTAA1* to sustain IAA levels

*OsAsp1* encodes an aspartic protease [24], implying that it may regulate the transcription of *OsTAA1* via an uncharacterised transcriptional regulator. As shown in our flowchart of *OsTAA1* transcription regulator prediction in rice (Fig 4A), we first used the *Arabidopsis* co-expression networks of *AtTAA1* (http://atted.jp/) [31] to identify a $C_2H_2$-type zinc-finger protein (At1G75710) as the candidate (S6A Fig), and then performed a BLAST search of the Rice Annotation Project (RAP) rice protein database (http://rapdb.dna.affrc.go.jp/) to obtain five homologs of At1G75710 that were presumed to regulate *OsTAA1* expression (S6B Fig). Based on their sequence homology and high expression in pistils, seeds, and endosperm (S6C Fig),

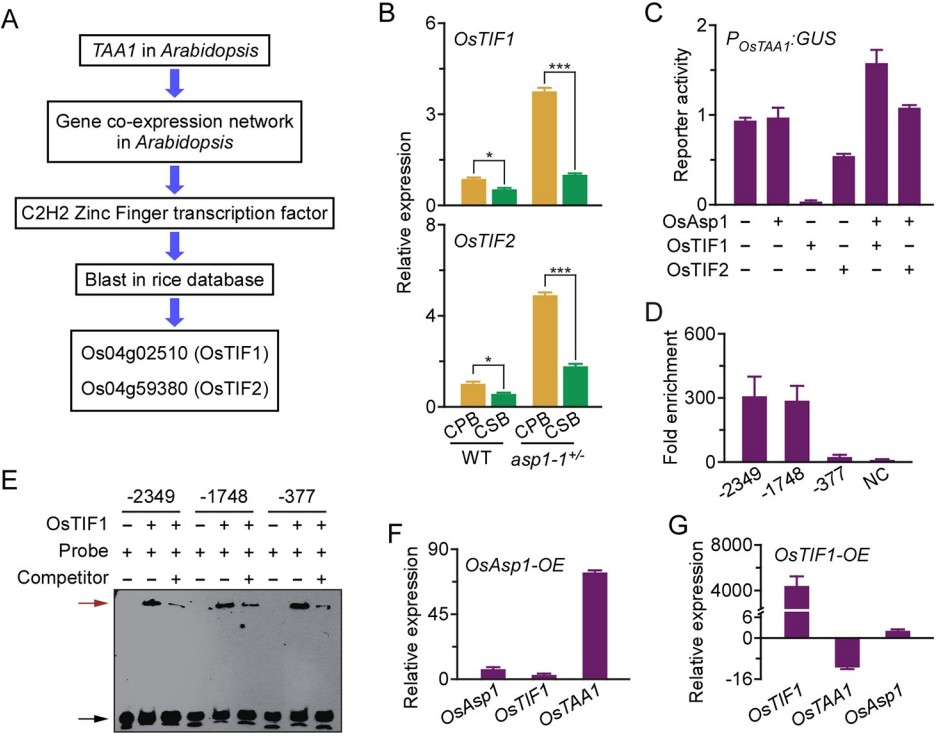

**Fig 4. OsAsp1 dismisses OsTIF1 binding to the *OsTAA1* promoter to regulate the transcription of *OsTAA1*. A**, Bioinformatics flowchart for the prediction of OsTAA1 transcription inhibition factors (OsTIF1 and OsTIF2) in rice. **B**, Transcript levels of *OsTIF1* and *OsTIF2* in CPBs and CSBs of the WT and *asp1-1^{+/−}* at 5 DAF. Target gene expression was normalised to that of *OsActin1*, and relative expression of the target gene in CPBs of the WT was set at 1.0. Values are means ± SD of three independent experiments. *$0.01 < P < 0.5$; ***$P < 0.001$ (Student's *t*-test). **C**, Promoter activity of $P_{OsTAA1}$ under the regulation of OsAsp1 and/or OsTIFs in the protoplast of rice leaves. The $P_{OsTAA1}$:GUS construct was transfected along with $P_{UBI}$:OsAsp1, $P_{UBI}$:OsTIFs, or $P_{UBI}$:OsAsp1 and $P_{UBI}$:OsTIFs, or with empty vector as control. We included *35S:LUC* as an internal control. Relative GUS activity normalised to luciferase activity was adopted as the promoter activity of $P_{OsTAA1}$. Values are means ± SD of three independent experiments. **D**, Chromatin immunoprecipitation (ChIP)–qPCR was used to detect the enrichment of OsTIF1-binding motifs at positions −2349, −1748, or −377 nt in the promoter region of *OsTAA1*. OsTIF1-HA was overexpressed in rice calli and purified with HA antibody. NC, negative control. Values are means ± SD of three independent experiments. **E**, EMSA showing recombinant OsTIF1-His directly binding the promoter region of *OsTAA1* at its potential motifs. The predicted OsTIF1 binding motifs at positions −2349, −1748, or −377 nt were labelled with biotin as the probe, and the unlabelled DNA fragments were used as the competitor. Red and black arrows indicate the shifted band and free probe, respectively. **F**, and **G**, Transcript levels of *OsAsp1*, *OsTIF1*, and *OsTAA1* in rice calli overexpressing *OsAsp1-GFP* (**F**) or *OsTIF1-HA* (**G**). The expression of each target gene was normalised to that of *OsActin1*, and calli transformed with empty vector were used as a control. qRT-PCR data are means ± SD of three independent experiments.

we identified two candidate genes, Os04g59380 and Os04g02510, which we named *OsTAA1 Transcription Inhibition Factor 1* (*OsTIF1*) and *OsTIF2*, respectively. We further detected the transcription levels of *OsTIF1* and *OsTIF2* in CPBs and CSBs of the WT and *asp1-1*$^{+/−}$, and found that expression of both genes was higher in both CPBs and CSBs of *asp1-1*$^{+/−}$than in the WT (Fig 4B). We transiently co-transformed $P_{OsTAA1}$:*GUS* with either OsAsp1 and OsTIF1 or OsTIF2 (S7A Fig) in rice mesophyll protoplasts, and found that both OsTIF1 and OsTIF2 inhibited the transcription activity of $P_{OsTAA1}$, and OsTIF1 more strongly (Fig 4C). In addition, OsAsp1 relieved the inhibitory effects of OsTIF1 or OsTIF2 on the activity of $P_{OsTAA1}$ in rice mesophyll protoplasts (Fig 4C).

Next, according to the target sequences of $C_2H_2$-type transcription factors reported previously [32], three potential OsTIF1 binding sites at −2349, −1748, and −377 nt were predicted in the 5' region upstream from the ATG site of *OsTAA1* (S7B Fig). Chromatin immunoprecipitation (ChIP)−qPCR showed that sites at positions −2349 and −1748 nt, but not at −377 nt, were enriched by OsTIF1 in transgenic rice calli (Fig 4D). An electrophoretic mobility shift assay (EMSA) showed that recombination OsTIF1-His could separately bind these three sites to produce the band shift visible in Fig 4E. We also found that overexpression of *OsAsp1* effectively upregulated transcription of *OsTAA1* (Fig 4F), whereas overexpression of *OsTIF1* strongly downregulated expression of *OsTAA1* in rice calli (Fig 4G). However, overexpression of *OsAsp1* and *OsTIF1* in rice calli slightly activated transcription of *OsTIF1* and *OsAsp1*, respectively (Fig 4F and 4G). These results demonstrate that OsAsp1 activates *OsTAA1* transcription by relieving the inhibition of OsTIF1 to ensure IAA biosynthesis.

## IAA induced OsTIF1 ER-nucleus transfer by reducing its interaction with OsAsp1 to form a feedback loop

We further investigated the possibility of an interaction between OsAsp1 and OsTIF1 to characterise the regulation mechanism of *OsTAA1* transcription. First, after separately co-transforming OsAsp1-GFP with an ER marker, OsTIF1-GFP with an ER marker, and OsAsp1-GFP with OsTIF1-mCherry, we showed that OsAsp1 and OsTIF1 were co-localised in the ER of rice mesophyll protoplasts (Fig 5A), which raised the question of how OsTIF1 enters the nucleus to inhibit *OsTAA1* expression. Due to the involvement of OsTAA1 in IAA formation in rice caryopses, we treated rice mesophyll protoplasts transformed with OsAsp1 and/or OsTIF1 with 50 μm IAA, and found that OsTIF1 was transferred into the nucleus, but that OsAsp1 remained localised in the ER after IAA treatment (Fig 5B). Then, using a yeast two-hybrid assay, we showed that OsAsp1 and OsTIF1 separately formed homodimers and interacted with each other (Fig 5C). Furthermore, IAA inhibited the interaction between OsAsp1 and OsTIF1 in the yeast two-hybrid system (Fig 5C). A co-immunoprecipitation (Co-IP) assay also demonstrated that OsAsp1-GFP interacted with OsTIF1-HA in rice mesophyll protoplasts and that their interaction was significantly depressed upon the addition of IAA (Fig 5D). These results indicate that IAA mediates a regulatory feedback loop involving the OsAsp1– OsTIF1 complex to fine-tune IAA levels between CPBs and CSBs during caryopsis development.

## *OsAsp1* transcription was regulated by auxin-inducible MADS29 after pollination

Our analyses showed that IAA levels were always higher in CPBs than in CSBs from the day of pollination (Fig 3A), which is consistent with the phenomenon of auxin apical dominance in plants. Therefore, we hypothesised that IAA regulates *OsAsp1* expression to maintain its differential level between CPBs and CSBs in rice. To test this hypothesis, we first cultured

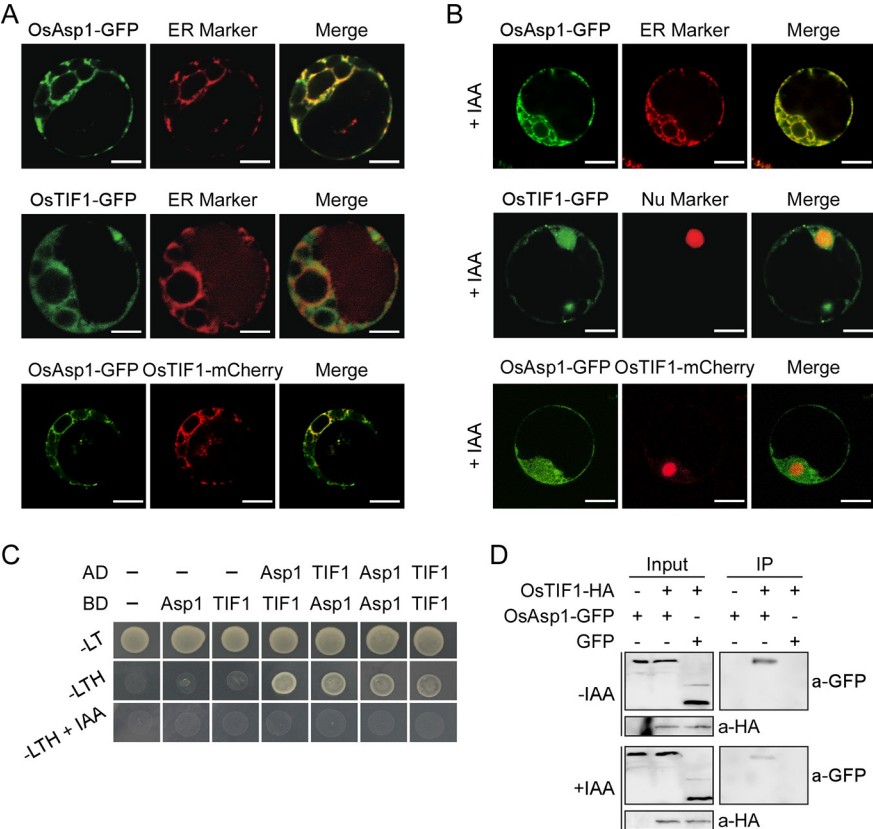

**Fig 5. OsTIF1 transfers from the endoplasmic reticulum (ER) into the nucleus after IAA releases its interaction with OsAsp1. A**, OsAsp1 co-localised with OsTIF1 in the ER of rice mesophyll protoplast. The ER marker was ER-rk CD3-959, purchased from the Arabidopsis Stock Center (http://www.arabidopsis.org/). Scale bars, 5 μm. **B**, OsTIF1, but not OsAsp1, was transferred from the ER into the nucleus in rice mesophyll protoplasts in response to IAA treatment (50 μM). The nuclear marker was AtBZR2(1–158 aa)-mCherry. Scale bars, 5 μm. **C**, Interaction between OsAsp1 and OsTIF1 detected by yeast two-hybrid assay.–LH indicates SD medium without LEU and TRP, and–LTH indicates SD medium without LEU, TRP, and HIS. **D**, Interaction between OsAsp1 and OsTIF1 detected by a Co-IP assay. $P_{UBI}$:*OsTIF1-HA* was co-transformed with $P_{UBI}$:*OsAsp1-GFP* or $P_{UBI}$:*GFP* into rice mesophyll protoplast, and then IAA (50 μM) was added, along with OsTIF1-HA as bait to pull down OsAsp1-GFP. Anti-GFP (a-GFP) was used to detect OsAsp1-GFP or green fluorescent protein (GFP), and anti- hemagglutinin (a-HA) was used to detect OsTIF1-HA.

unpollinated spikelets (-1 DAF) *in vitro* with IAA, and found that transcription of *OsAsp1* was induced in a concentration-dependent manner (Fig 6A). In addition, *in vitro* culture of unpollinated spikelets with 2,4-dichlorophenoxyacetic acid (2,4-D) also induced *OsAsp1* expression (S8 Fig). Moreover, *OsAsp1* transcription in spikelets of CSBs strongly increased after treatment with IAA at 3 days before fertilisation (-3 DAF) compared with *OsAsp1* transcription in spikelets given a mock treatment (methanol) (Fig 6B). The grain weight of mature CSB seeds after IAA treatment was also greater than that of CSB seeds treated with mock solution and similar to that of CPB seeds (Fig 6C and S9 Fig). We further treated CPBs of *asp1-1*$^{+/-}$with IAA to restore the gradient of IAA levels between CPBs and CSBs, and found that the grain weight of mature CPB seeds of *asp1-1*$^{+/-}$after IAA treatment was greater than that of CSB seeds (S10A Fig), and that this difference was similar to the grain weight difference between CPB and CSB seeds in the WT. In addition, the difference in *OsAsp1* expression between CPBs and CSBs in *asp1-1*$^{+/-}$ was also complemented by IAA treatment of CPBs in *asp1-1*$^{+/-}$ (S10B Fig).

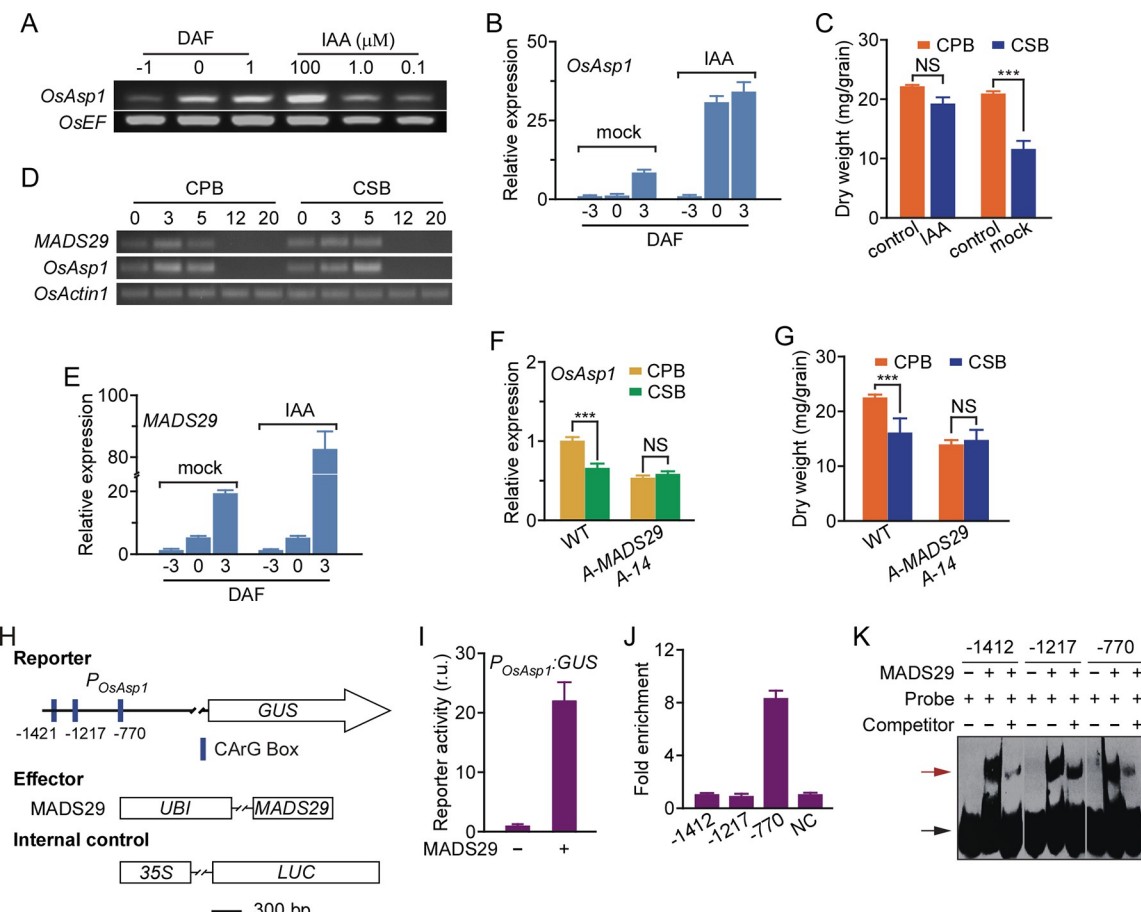

**Fig 6. IAA promotes the transcription of *OsAsp1* through MADS29 to maintain differential levels of OsAsp1 in CPBs and CSBs after pollination. A**, *OsAsp1* levels detected by semi-quantitative RT-PCR in spikelets at –1, 0, and 1 DAF, and in detached spikelets at –1 DAF after culture in Murashige and Skoog (MS) medium plus IAA for 24 h. *OsActin1* was used as a control. **B**, Quantification of *OsAsp1* levels in CSBs of the WT treated at –3 and 0 DAF with IAA, or with methanol as a mock treatment. *OsAsp1* expression was normalised to that of *OsActin1*, and relative expression of *OsAsp1* at –3 DAF in mock-treated samples was set at 1.0. Values are means ± SD of three independent experiments. **C**, Mature grain dry weight of CPBs and CSBs after the IAA treatment described in **B**. Values are means ± SD ($n = 30$). NS, no significant difference; ***$P < 0.001$ (Student's *t*-test). **D**, Transcript levels of *MADS29* and *OsAsp1* detected by RT-PCR in CPBs and CSBs of the WT at 0, 3, 5, 12, and 20 DAF. *OsActin1* was used as a control. **E**, Transcript level of *MADS29* in CSBs of the WT in response to IAA treatment. CSB spikelets on panicles were treated at –3 and 0 DAF with IAA, or with methanol as mock treatment. *MADS29* expresion was normalised to that of *OsActin1*, and relative expression of *MADS29* at –3 DAF in mock-treated samples was set at 1.0. Values are means ± SD of three independent experiments. **F**, Transcript levels of *OsAsp1* in CPBs and CSBs of the WT and *Antisense-MADS29* transgenic rice A14 (*A-MADS29 A-14*) at 5 DAF. *OsAsp1* expression was normalised to that of *OsActin1*, and relative expression in CPBs of the WT was set at 1.0. Values are means ± SD of three independent experiments. NS, no significant difference; ***$P < 0.001$ (Student's *t*-test). **G**, Mature grain dry weight of CPBs and CSBs of the WT and *A-MADS29 A-14*. Values are means ± SD ($n = 30$). ***$P < 0.001$ (Student's *t*-test); NS, no significant difference. **H**, Constructs used for GUS assay of *OsAsp1* promoter activity. The three predicted MADS29 binding sites (CArG box) were labelled in the 1500-bp promoter region of *OsAsp1*. *LUC*, *firefly luciferase*; *GUS*, *beta-glucuronidase*. Scale bars, 300 bp. **I**, Promoter activity of $P_{OsAsp1}$ activated by MADS29 in the protoplast of *Arabidopsis* leaves. $P_{OsAsp1}$:GUS was co-transfected with MADS29 or an empty vector as a control. *35S:LUC* was used as an internal control. Relative GUS activity normalised to luciferase activity was adopted as the promoter activity of $P_{OsAsp1}$. Values are means ± SD of three independent experiments. **J**, ChIP-qPCR was used to detect the enrichment of CArG boxes by MADS29 at sites –1421, –1217, and –770 of the *OsAsp1* promoter region. *35S:MADS29-GFP* and $P_{OsAsp1}$:GUS were co-transformed into the protoplast of *Arabidopsis* leaves, and MADS29-GFP was purified with GFP antibody. NC, negative control. Values are means ± SD of three independent experiments. **K**, EMSA showing the recombinant MADS29-His directly binding the promoter region of *OsAsp1* on its CArG boxes. The CArG boxes at positions –1421, –1217, and –770 were labelled with biotin as the probe, and the unlabelled DNA fragments were used as the competitor. Red and black arrows indicate the shifted band and free probe, respectively.

To further investigate the molecular mechanism regulating *OsAsp1* expression, we identified genes that were co-expressed with *OsAsp1* using the Rice Oligo Array Database [33], and found five MADS family members with enriched expression (S3 Table). A prior study showed that MADS29 is induced by IAA and is involved in rice seed development [34]. We found that the expression pattern of *MADS29* was similar to that of *OsAsp1* during the development of CPBs and CSBs after fertilisation (Fig 6D), and that IAA treatment of CSB spikelets at –3 DAF increased *MADS29* transcription compared to that in mock-treated spikelets (Fig 6E). Furthermore, *OsAsp1* expression was almost the same and downregulated in the CPBs and CSBs of *Antisense-MADS29* transgenic rice lines *A2* and *A14*, respectively, but not in *A33*. (Fig 6F and S11 Fig). The difference in mature grain dry weight between CPBs and CSBs was absent in *Antisense-MADS29* lines *A2* and *A14*, but not in *A33*, which was similar to that of *asp1-1*$^{+/-}$ (Fig 6G and S12 Fig). Three MADS protein binding sites, CArG boxes at positions –1412, –1217, and –770 nt were predicted in the 5' region upstream from the ATG site of *OsAsp1* (Fig 6H). We found that MADS29 effectively promoted the activity of $P_{OsAsp1}$:*GUS* in *Arabidopsis* mesophyll protoplasts (Fig 6I). ChIP–qPCR analysis showed that the CArG box at position –770 nt, but not those at positions –1412 or –1217 nt, was enriched by MADS29-His in transgenic rice calli (Fig 6J). An EMSA further showed that recombinant MADS29-His bound all three of these CArG boxes to produce a specific shifted band (Fig 6K). These results indicate that IAA promotes the expression of *OsAsp1* via MADS29 to sustain higher OsAsp1 levels in CPBs than in CSBs during pollination.

Based on these results, we propose a model to explain how OsAsp1 maintains auxin apical dominance to regulate the developmental differential between CPBs and CSBs (Fig 7). In this model, high IAA levels upregulate *OsAsp1* transcription via MADS29 at 0 DAF to produce higher *OsAsp1* levels in CPBs than in CSBs, which explains why *OsAsp1* was identified as differentially expressed in our transcriptome-wide expression analysis. Then, OsAsp1 interacts with OsTIF1 in the ER to dispel OsTIF1 transcriptional inhibition of *OsTAA1*, causing IAA content to peak at 5 DAF. At the same time, IAA induces transfer of OsTIF1 into the nucleus by reducing its interaction with OsAsp1, causing IAA biosynthesis to decrease, and thereby forming a feedback regulation loop controlling IAA levels. From this model, we conclude that differential IAA levels between CPBs and CSBs are determined by the OsAsp1–OsTIF1 complex, which is essential for the distinct development of CPBs and CSBs.

## Discussion

Auxin apical dominance has evolutionary advantages for plant growth and reproduction because it plays an essential role in regulating plant shoot architecture and ensuring the production of at least some high-quality seed within a short period [35, 36]. In this study, we demonstrated that differential distribution of auxin determines developmental differences between CPBs and CSBs in rice, causing the grain-filling problem observed in CSBs and thereby reducing total grain production and quality. We found that auxin-dependent transcription of *OsAsp1* and interaction of OsAsp1 with OsTIF1 regulate the transcription of *OsTAA1* to maintain auxin apical dominance between CPBs and CSBs after fertilisation, providing a potential strategy for enhancing grain size and grain filling and thereby increasing rice yield.

Asps comprise a large endopeptidase family that is widely distributed in all three domains of life [37, 38]. The rice genome contains 96 putative *OsAsp* genes [39], among which atypical and nucellin-like Asps are characterised by the absence of a plant-specific insert and an unusually high number of cysteine residues, which play regulatory roles in biotic and abiotic stress responses, chloroplast metabolism, and reproductive development [40]. A prior study showed that EAT1 (ETERNAL TAPETUM 1), a basic helix-loop-helix transcription factor, promotes

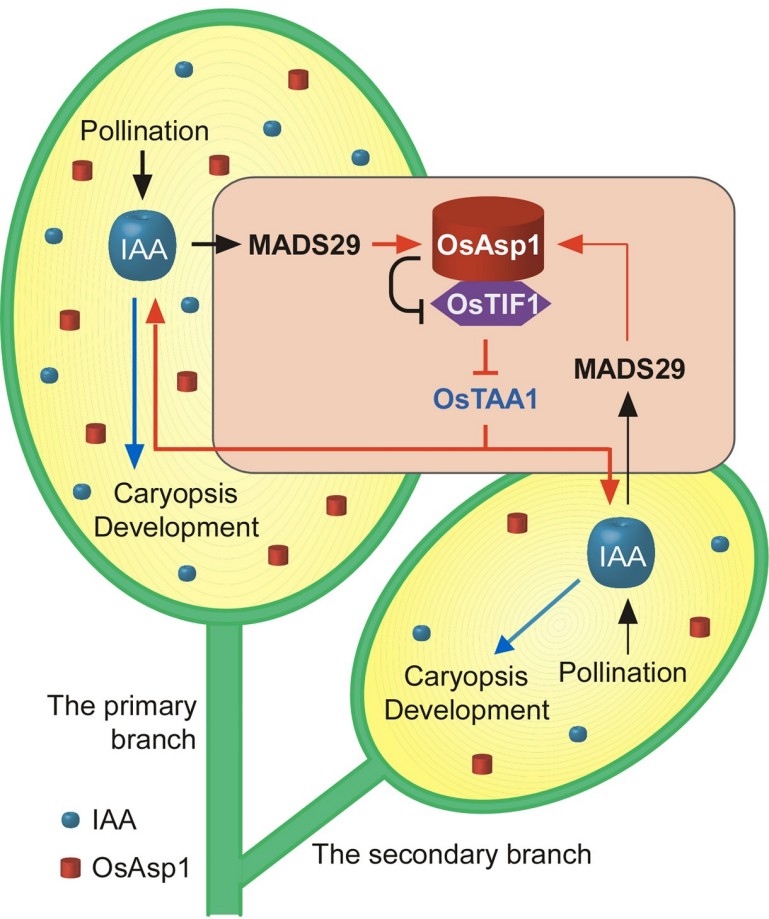

**Fig 7. Schematic model of the OsAsp1–OsTIF1 complex maintaining IAA apical dominance to establish differential development between CPBs and CSBs in rice.**

tapetal programmed cell death (PCD) by directly regulating the expression of *OsAsp25* and *OsAsp37* in rice anthers [41]. However, caspase activity of OsAsp25 or OsAsp37 was not detected, indicating that OsAsp25 and OsAsp37 regulate downstream caspase-like protease(s) involved in PCD rather than directly acting on caspase-specific substrates [41]. Rice nucellin gene *OsAsp1* is strongly expressed in zygotic embryos at 1–2 DAF [24]. However, the biological function of OsAsp1 remains unclear. Auxin has been found to induce OsMADS29 transcription to promote PCD of the nucellus and nucellar projection during rice seed development [34]. The present study was the first to determine that OsAsp1 is essential for rice caryopse development because both the *asp1-1* and *asp1-2* T-DNA insertion mutant lines were homozygous-lethal and, compared to the WT, the difference in mature seed grain weight between CPBs and CSBs disappeared in both lines. Furthermore, we showed that auxin regulates expression of *OsAsp1* through OsMADS29 to maintain high levels of OsAsp1 in CPBs, but not in CSBs, after fertilisation, and that OsAsp1 then interacts with OsTIF1 and dismisses its transcriptional inhibition for OsTAA1 to promote IAA biosynthesis. In addition, IAA facilitates OsTIF1 translocation from the ER to the nucleus by releasing the interaction of OsTIF1 with OsAsp1 to form a feedback regulation of IAA levels in caryopses. The disruption of *OsAsp1* levels in *asp1-1$^{+/-}$* and overexpression of *OsAsp1* in rice calli activates transcription of *OsTIF1*, indicating that *OsTIF1* expression is regulated by an uncharacterised and

sophisticated mechanism. Therefore, our results demonstrate that OsAsp1 functions as a transcriptional regulator to form a complex with OsTIF1 to regulate the differential development of CPBs and CSBs after fertilisation.

As a critical phytohormone, auxin regulates nearly all aspects of plant development including morphogenesis and adaptive responses [42]. Auxin distribution depends on both auxin metabolism (biosynthesis, conjugation, and degradation) and cellular auxin transport [43, 44]. Auxin is mainly synthesised in shoot apical meristem to maintain increased shoot apical dominance, which transfers to inflorescence meristem when plants flower. In higher plants such as *Arabidopsis*, maize, and rice, the IPA pathway, catalysed by TAA and YUCCA, has been shown to be the major auxin biosynthetic pathway [29, 30, 45–47]. The rice genome contains two *OsTAA* (synonym of *OsTAR*) and 14 *OsYUCCA* genes; increased IAA content during development of rice grains was strongly correlated with the expression of *OsTAR1*, *OsYUCCA9*, and *OsYUCCA11* [48]. IAA content has been shown to be higher in CPBs than in CSBs at the early grain-filling stage [16], which is consistent with our results. However, few studies have focused on the differential expression of *OsTAAs* and *OsYUCCAs* in CPBs and CSBs. In this study, we showed that *OsTAA1*/*OsTAR2* expression gradually increased after fertilisation, reaching a peak at 5 DAF in both CPBs and CSBs in the WT, and with a higher peak in CPBs, which was correlated with the IAA content of CPBs and CSBs in the WT. However, in *asp1-1*$^{+/-}$, expression peaks of *OsTAA1* disappeared at 5 DAF, but occurred at 3 DAF in both CPBs and CSBs, indicating that *OsTAA1* transcription is regulated by OsAsp1 and that OsTAA1 is a key enzyme for IAA biosynthesis, maintaining differential auxin distribution between CPBs and CSBs. By contrast, the expression pattern of *OsYUCCA1*, which was more highly expressed in CPBs than in CSBs at 5 and 10 DAF, respectively, was similar in the WT and *asp1-1*$^{+/-}$, except that *OsYUCCA1* expression was lower in CPBs in *asp1-1*$^{+/-}$ than in the WT at 10 DAF. These results imply that OsYUCCA1 is not a main enzyme for IAA biosynthesis in CPBs and CSBs. Therefore, we will aim to detect the expression levels of 13 other *OsYUCCAs* of CPBs and CSBs in *asp1-1*$^{+/-}$ in a future study to determine which *OsYUCCA* is regulated by OsAsp1 and associated with IAA biosynthesis in CPBs and CSBs. The heterologous expression of *OsAsp1* in *Arabidopsis* improved IAA levels in root apical meristem in 3-day-old seedlings, as well as transcription levels of *AtTAA1* and *AtYUCCA8*, but not *AtYUCCA9*, providing additional support for the detection of expression of other *OsYUCCAs* in *asp1-1*$^{+/-}$.

In this study, we characterised the effect of differential auxin distribution mediation by OsAsp1 on the developmental difference between CPBs and CSBs. First, we found that two independent T-DNA insertion mutant lines, *asp1-1* and *asp1-2*, present a homozygous-lethal phenotype and that differences in ovary development and size, endosperm cellularisation and development, and mature seed grain weight between CPBs and CSBs disappeared, and were closer to CPBs in *asp1-1*$^{+/-}$. These results indicate that OsAsp1 is critical for grain development and maintenance of the developmental difference between CPBs and CSBs. We further showed that IAA content was very low and almost identical in CPBs and CSBs in *asp1-1*$^{+/-}$ during the early stages of caryopse development, indicating that low auxin levels are sufficient to trigger embryo and endosperm development in CPBs and CSBs, and promote similar development of CSBs and CPBs in *asp1-1*$^{+/-}$. *In vitro* treatment of CPBs in *asp1-1*$^{+/-}$ with IAA phenocopied the expression difference of *OsAsp1* and the development difference between CPBs and CSBs in the WT. These results imply that the higher auxin levels in CPBs than in CSBs, but not absolute IAA concentration, are critical for maintaining differential development of CPBs and CSBs. Treatment of CSBs in the WT with extra IAA before pollination promoted the development of CSBs, allowing the grain weight of CSBs to increase to that of CPBs, which provides strong evidence for differential IAA distribution, but not concentration, determining

the developmental difference between CPBs and CSBs in rice. Auxin is produced post-fertilisation in seeds, which drives central cell division and is necessary for MADS-box transcription factor AGL62 (AGAMOUS-LIKE 62)-dependent endosperm development in *Arabidopsis* [28]. In the present study, we showed that auxin upregulates *OsAsp1* transcription via MADS29 after fertilisation to produce higher levels of *OsAsp1* in CPBs than in CSBs. The interaction of OsAsp1–OsTIF1 regulated by auxin formed a feedback regulation loop to maintain auxin apical dominance during rice caryopsis development. Therefore, our results present a mechanistic explanation for differential distribution of auxin between CPBs and CSBs to regulate distinct caryopse development patterns in rice.

## Materials and methods

### Plant materials and growth conditions

The WT rice (*Oryza sativa* L. subspecies *japonica*) varieties used in this study were Zhong Hua 15, for RNA sequencing (RNA-seq) and auxin treatment; Zhong Hua 11 for *A-MADS29* lines; and Dong Jin for *asp1-1* and *asp1-2*. Two T-DNA mutant lines, PFG_3A-60042 and PFG_3A-04889, were obtained from http://signal.salk.edu, named *asp1-1* and *asp1-2*, respectively, and confirmed by genomic PCR and RT-PCR. Both rice lines were cultivated in a field at Beijing Normal University during the natural growing season for observation of their growth phenotypes during the vegetative and reproductive periods.

### RNA-seq and data analysis

Caryopses were collected from growing Zhong Hua 15 rice at different stages after heading for total RNA extraction. Libraries for each sample were constructed using a Next Multiplex RNA Library Prep Set (NEB) and sequenced on an Illumina HiSeq 2000 Platform. Clean reads were mapped to the rice genome (version 7.0) [49] using Cufflink software (http://cufflinks.cbcb. umd.edu/). For gene expression analysis, the numbers of reads per kilobase per million mapped reads (RPKM) were calculated. Differentially expressed genes were defined as having a fold change of $\geq 2$ or $\leq 0.5$ and a false discovery rate (FDR) $\leq 0.05$.

### qRT-PCR and semi-quantitative RT-PCR

Total RNA was separately extracted from caryopses at different developmental stages, transgenic rice calli, and 5-day-old *Arabidopsis* seedlings using TRIzol reagent (Invitrogen, USA) and purified using a PureLink RNA Mini Kit (Invitrogen) and a PureLink DNase Kit (Invitrogen), according to the manufacturer's protocol. Approximately 2 μg RNA was reverse transcribed using a Reverse-Aid First Strand cDNA Synthesis Kit (Thermo Fisher Scientific, USA) and used as template for RT-PCR and RT-qPCR. PCR reactions were carried out in a MyCycler thermal cycler (Bio-Rad, USA) using extension cycling conditions of 94˚C for 30 s, 55˚C for 30 s, and 72˚C for 30 s. The amplifications were performed for 28–33 cycles. qRT-PCR was performed using a 7500 Fast Real-Time PCR System (Applied Biosystems, USA) with Power SYBR Green PCR Master Mix (Applied Biosystems). The thermal program was 2 min at 50˚C, 10 min at 95˚C, followed by 40 cycles of 15 s at 95˚C and 60 s at 60˚C. The dissociation curve program was used to confirm the specificity of the target amplification product. Three independent biological replicates were performed for RT-qPCR analyses. *OsActin1* or *AtUBQ10* was used as an internal control in both qRT-PCR and RT-PCR analyses. The primers for the target genes are listed in S4 Table.

## Eosin staining

Whole-mount stain-clearing laser-scanning confocal microscopy (WCLSM) was used to evaluate the structure of mature embryo sacs of WT and *asp1*$^{+/-}$ mutant rice. The spikelets of WT and *asp1*$^{+/-}$ mutants were collected, fixed, hydrated, stained in Eosin Y water solution, dehydrated, and cleared as previously described [50]. The developmental phenotype of the embryo sacs was examined using confocal laser-scanning microscopy (LSM 700, Zeiss, Germany).

## Flow cytometry analysis

Ovaries of WT and *asp1-1*$^{+/-}$ rice sampled at 3 or 5 DAF were chopped with a razor blade (six ovaries per group). We added 500 μL GS buffer (45 mM $MgCl_2$, 30 mM sodium citrate, 20 mM MOPS, 0.1 v/v Triton X-100) to each group and the mixture was filtered by passing it through a sieve (48 μm$^2$ mesh). DNA was stained by adding 10 mM propidium iodide. The DNA content of the nuclei was measured using a flow cytometer (BD FACSAria II; BD Biosciences) as previously reported [51].

## Liquid chromatography–mass spectrometry (LC-MS) analysis

IAA content was quantified using LC–MS as previously reported [52]. Fresh caryopses were collected and ground into powder using liquid nitrogen. Approximately 100 ± 5 mg powder was extracted using 1.5 mL methanol. The supernatant was lyophilised in a freeze-drier (Thermo Fisher Scientific) and then resuspended in 85% (v/v) methanol in water for LC–MS analysis. IAA (Sigma-Aldrich) was used as a standard to prepare the standard curve.

## EMSA analysis

The coding sequences (CDSs) of *MADS29* and *OsTIF1* were separately amplified and cloned into the vector pET32a. MADS29-His and OsTIF1-His were expressed in *Escherichia coli* (BL21) and affinity-purified using Ni-NTA Resin. Synthesised fragments of the DNA probes (S4 Table) were labelled using a Biotin 3'-End DNA Labeling Kit according to the manufacturer's instructions (Thermo Fisher Scientific). The protein–DNA binding reactions were performed using a LightShift Chemiluminescent EMSA Kit according to the manufacturer's instructions (Thermo Fisher Scientific).

## ChIP-qPCR

*Ubi1*:*OsTIF1-HA* and *Ubi1*:*MADS29-GFP* were transformed into rice callus by particle bombardment using a PDS-1000/He biolistic particle delivery system (Bio-Rad) as described previously [53]. ChIP of OsTIF1-HA and MADS29-GFP was performed as previously described, with minor modifications [54]. Briefly, 1 g transgenic callus was ground to a fine powder with liquid nitrogen and combined with 10 mL ChIP extraction buffer (0.4 M sucrose, 10 mM Tris-HCl, 10 mM $MgCl_2$, 1 mM DTT, 0.1 mM PMSF and protease inhibitor cocktail, pH 8.0). Then, 270 μL 37% formaldehyde solution (to a final concentration of 1%) was added and the mixture was incubated at 4°C for 10 min to cross-link DNA to protein. The cross-linking reaction was quenched by adding 0.63 mL 2 M glycine, and the mixture was incubated at 4°C for 5 min. Finally, the nuclear pellet was isolated by centrifugation and resuspended with ChIP lysis buffer (50 mM Tris-HCl, 10 mM EDTA, and 1% sodium dodecyl sulfate [SDS], pH 8.0) and kept on ice for 30 min. One volume of ChIP dilution buffer (16.7 mM Tris-HCl, 167 mM NaCl and 1.1% Triton X-100, pH 8.0) was added to the samples before sonication for 12 min (30 s on, 30 s off, high level) in a Bioruptor (Diagenode) to yield DNA fragments of 0.2–1.0 kb in length. The lysates were diluted fivefold in ChIP dilution buffer to decrease the concentration

of SDS to 0.1% and cleared by centrifugation ($16,000 \times g$ for 5 min at 4˚C). After keeping 5% of the sample as an input control, the remaining supernatant was incubated with antibody-bound Dynabeads Protein A or G (Invitrogen) overnight at 4˚C. Washing, elution, reverse cross-linking, and DNA purification were performed according to the methods of a previous study [55]. The antibodies used for ChIP were hemagglutinin (HA, Sigma) and green fluorescent protein (GFP, Sigma). DNA isolated by ChIP was used for qPCR analysis. The 100-bp 5'-region of *OsTAA1* was used as a negative control for OsTIF1, and the 100-bp N-terminal region of GUS was used as a negative control for MADS29. The primers used for qPCR are listed in S4 Table.

## Transient transactivation assay

The promoters of *OsTAA1* or *OsAsp1* were amplified by PCR from rice genomic DNA and cloned into the pGPTV vector to generate the $P_{OsTAA1}$:*GUS* and $P_{OsAsp1}$:*GUS* constructs, respectively. $P_{OsTAA1}$:*GUS* was co-transformed with either *35S:Luc* and *35S:OsAsp1-GFP* or *35S:OsTIF1-HA* into rice mesophyll protoplasts [56, 57]. $P_{OsAsp1}$:*GUS* was co-transformed with *35S:Luc* and *35S:MADS29-GFP* into *Arabidopsis* mesophyll protoplasts [58]. *35S:Luc* was used as an internal control. GUS and luciferase (Luc) activities were measured as previously described [59], and relative GUS activity (GUS/Luc) was calculated to determine the activity of $P_{OsTAA1}$ or $P_{OsAsp1}$.

## Yeast two-hybrid assay

The CDS of *OsAsp1* or *OsTIF1* was subcloned into pGAD7 or pGBK7 (Clontech) as prey and bait constructs, respectively. Then, the pairs of bait and prey constructs were co-transformed into yeast strain AH109 and screened by growth on selective medium (SD–Leu–Trp–His). The same strains were grown on a control medium (SD–Leu–Trp) for use as controls.

## Co-IP assays

*Ubi1:OsTIF1-HA* was co-transformed with *Ubi1:OsAsp1-GFP* or *Ubi1:GFP* into rice mesophyll protoplasts, which then were incubated with or without 50 µM IAA for 16 h. Total protein was extracted and immunoprecipitated with GPF or HA antibody incubated with antibody-bound Dynabeads Protein A or G. The immunoprecipitates were separated by 10% SDS–polyacrylamide gel electrophoresis (PAGE) and detected with GFP antibody. *Ubi1:HA* and *Ubi1:OsAsp1-GFP* transgenic protoplasts were used as a negative control.

## IAA and 2,4-D treatments

Spikelets were collected from panicles at -1 DAF and cultured in Murashige and Skoog (MS) medium plus IAA at different concentrations (100, 1.0, or 0.1 µM) or 2,4-dichlorophenoxyacetic acid (2,4-D, Sigma) at different concentrations (10, 1.0, or 0.1 µM) in 0.1 v/v Triton X-100. Some of the collected spikelets were cultured in the MS medium with methanol in 0.1 v/v Triton X-100 as a control, as previously described [34, 60]. Spikelets located on CSBs in the WT and on CPBs in *asp1-1*$^{+/-}$ were separately treated with 100 µM IAA (Sigma) in 0.1 v/v Triton X-100 using a fine brush once per day from –3 to 0 DAF. Methanol treatment used as a negative control (mock), following a method modified from a previous study [13].

## Generation of transgenic *Arabidopsis* plants and GUS staining

The constructs *35S:OsAsp1-GFP* and *DR5:GUS* [61] were separately introduced into *Agrobacterium* GV3101, and transformed into *Arabidopsis* WT (Col-0) using the floral dip

transformation method [62]. The transformants were screened with 1/2 MS medium containing 50 mg/L hygromycin B (Amresco) and 50 mg/L kanamycin (INALCO) to obtain T3-generation homozygous transgenic lines. *DR5:GUS* transgenic lines were crossed with *OsAsp1-overexpressed* lines to generate *DR5:GUS x OEOsAsp1-GFP* plants for GUS staining The 3-day-old seedlings were stained in the GUS stain buffer (0.5 mg/mL X-Gluc, 0.01 v/v TritonX-100, 0.01 v/v DMSO, 10 mmol/L EDTA in sodium phosphate buffer, pH 7.0) for 2 h at 37˚C, and then destained in ethanol solution for photographing with confocal laser-scanning microscopy (LSM 700, Zeiss).

## Supporting information

**S1 Table. Candidate genes were more highly expressed in apical primary branches (CPBs) at 0 days after heading (DAH) than in 5-DAH proximal secondary branches (CSBs), and expression was higher in 12-DAH CSBs than in 5-DAH CPBs.**
(DOCX)

**S2 Table. Genotype of the F1 progeny of two *asp1* independent lines self-crossed or back-crossed with wild-type (WT) plants.**
(DOCX)

**S3 Table. Co-expression genes of *OsAsp1* from the Rice Oligo Array Database (ROAD).**
(DOCX)

**S4 Table. Primers and probe sequences used in the study.**
(DOCX)

**S1 Fig. Development status of apical primary branches (CPBs) and proximal secondary branches (CSBs) at different days after heading (DAH). A**, Developmental phenotypes of CPBs and CSBs during the period after heading. Scale bars, 20 μm. **B**, Cell proliferation rates of CPBs and CSBs during the period after heading. **C**, Filling rates of CPBs and CSBs during the period after heading. Arrows indicate the highest filling rates. Red and blue lines indicate CPBs and CSBs, respectively.
(JPG)

**S2 Fig. Transcription levels of *Os11g08200* (*OsAsp1*), *Os07g39020* and *Os06g41030* in CPBs and CSBs detected by quantitative reverse-transcription polymerase chain reaction (qRT-PCR) at different DAH.** RNA samples were extracted from caryopses at eight development stages (CPB-0, -5, -12, -20 and CSB-5, -12, -25, -35). The expression of each target gene was normalised to that of *OsActin1*, and the relative expression of CPB-0 was set to 1.0. Data are means ± standard deviation (SD) of three biological replicates.
(JPG)

**S3 Fig. Growth phenotypes of the rice T-DNA insertion mutant lines *OsAsp1-1*$^{+/-}$ and *OsAsp1-2*$^{+/-}$. A**, Plant height of rice wild-type (WT) and mutant lines *asp1-1*$^{+/-}$ and *asp1-2*$^{+/-}$ at 40 DAH. **B**, Spikelet numbers per panicle in the WT, *asp1-1*$^{+/-}$, and *asp1-2*$^{+/-}$. Values are means ± SD ($n$ = 10). *$0.01 < P < 0.5$ (Student's *t*-test). NS, no significant difference.
(JPG)

**S4 Fig. Mature grain phenotypes of CPBs and CSBs in the WT and mutant lines *asp1-1*$^{+/-}$ and *asp1-2*$^{+/-}$.** The developed grains were collected at 40 DAH from CPBs and CSBs in all plants.
(JPG)

**S5 Fig. Ovary length and width of CPBs and CSBs in the WT and *asp1-1*$^{+/-}$ at 3 and 5 days after flowering (DAF). A**, Ovary length. **B**, Ovary width. Values are means ± SD (*n* = 6). NS, no significant difference; $^{*}0.01 < P < 0.5$; $^{***}P < 0.001$ (Student's *t*-test).
(JPG)

**S6 Fig. Potential regulation factors of *OsTAA1* predicted by the *Arabidopsis* co-expression network. A**, Genes co-expressed with *AtTAA1* (*At1g70560*, *WEI8*) were forecast from an *Arabidopsis* gene co-expression network (http://atted.jp/). Red rectangle indicates the potential transcription factor (At1g75710, a zinc-finger protein) of *AtTAA1*. **B**, Genes homologous to *At1g75710* were obtained by BLAST searching of the rice genome sequence at http://rapdb.dna.affrc.go.jp/. **C,** Expression profiles of five rice homologous genes of At1g75710 in different tissues and organs. The RNA sequencing (RNA-seq) expression values of these five genes (in fragments per kb per million mapped reads, FPKM) were extracted from http://rice.plantbiology.msu.edu/index.shtml. Gene expression values are presented as a heat map.
(JPG)

**S7 Fig. Constructs for the GUS assay of promoter activity of *OsTAA1* and predicted OsTIF1 binding sites located in the promoter region of *OsAsp1*. A**, Constructs used for GUS assays. *LUC, firefly luciferase*; *GUS, beta-glucuronidase*. Scale bars, 800 bp. **B**, OsTIF1 binding sites located in the 2500 bp upstream from the ATG site of *OsTAA1*. Red, blue, and green boxes indicate the OsTIF1 binding sites. The 30-bp sequence around the binding site was used as a probe for the electrophoretic mobility shift (EMSA) assay (Fig 2E). Scale bars, 300 bp.
(JPG)

**S8 Fig. Transcription levels of OsAsp1 detected by semi-quantitate RT-PCR in spikelets at -1, 0 and 1 DAF, and in detached spikelets at -1 DAF, after culture in Murashige and Skoog (MS) medium plus 2,4-D for 24 h. OsActin1 was used as a control.**
(TIF)

**S9 Fig. Mature grain phenotypes of CPBs and CSBs in the WT in response to IAA treatment. CSBs were treated with IAA at each time point from -3 to 0 DAF, or with methanol as a control (mock).** CPBs without treatment were used as a control. Scale bars, 10 mm.
(JPG)

**S10 Fig. Mature grain weight and expression of *OsAsp1* in CPBs and CSBs in one whole plant of *asp1-1*$^{+/-}$ following IAA treatment. A**, Mature grain dry weight of CPBs and CSBs for *asp1-1*$^{+/-}$ with IAA treatment. CPBs were treated with IAA or methanol as a control (mock). CSBs without treatment were used as a control check. Values are means ± SD (*n* = 30). Different letters indicate significant differences at $P < 0.05$ (Student's *t*-test). **B**, Transcription levels of *OsAsp1* were detected by qRT-PCR in CPBs and CSBs in *asp1-1*$^{+/-}$ with the same IAA treatment as (**A**) at 3 DAF. *OsAsp1* expression was normalised to that of *OsActin1*, and the relative expression of CSBs was set at 1.0. Values are means ± SD of three independent experiments. NS, no significant difference; $^{***}P < 0.001$ (Student's *t*-test).
(JPG)

**S11 Fig. Transcription levels of *OsAsp1* of CPBs and CSBs in the WT and *Antisense-MADS29* (*A-MADS29*) lines A-2 and A-33 at 5 DAF were detected by qRT-PCR.** The expression of *OsAsp1* was normalised to that of *OsActin1*, and relative expression in CPBs in the WT was set at 1.0. Values are means ± SD of three independent experiments. NS, no significant difference; $^{*}0.01 < P < 0.5$; $^{***}P < 0.001$ (Student's *t*-test).
(JPG)

**S12 Fig. Mature grain phenotype of CPBs and CSBs in the WT and *Antisense-MADS29* (*A-MADS29*) lines. A**, Mature grains of CPBs and CSBs in the WT and *A-MADS29* A-14 were collected at 40 DAH. Scale bars, 10 mm. **B**, Developed grains of CPBs and CSBs in the WT, *A-MADS29* A-2, and A-33 were collected at 40 DAH. Scale bars, 5 mm. **C**, Mature grain dry weight of CPBs and CSBs in the WT and *A-MADS29 A-2* and *A-33* as in (**B**). Values are means ± SD ($n = 30$). $^{***}P < 0.001$ (Student's *t*-test); NS, no significant difference. (JPG)

## Acknowledgments

We thank Dr. Hongwei Xue for providing the antisense-*MADS29* transgenic lines A-2, A-14, and A-33. We also acknowledge the Experimental Technology Center for Life Sciences, Beijing Normal University for experimental equipment utilization.

## Author Contributions

**Data curation:** Shu Chang, Yixing Chen, Yihao Li, Shengcheng Han, Yingdian Wang.

**Funding acquisition:** Shengcheng Han, Yingdian Wang.

**Methodology:** Shu Chang, Yixing Chen, Shenghua Jia, Kun Liu, Zhouhua Lin, Hanmeng Wang, Zhilin Chu, Shengcheng Han, Yingdian Wang.

**Software:** Jin Liu, Chao Xi.

**Supervision:** Shengcheng Han, Yingdian Wang.

**Visualization:** Jin Liu, Chao Xi.

**Writing – original draft:** Shu Chang, Shengcheng Han, Yingdian Wang.

**Writing – review & editing:** Heping Zhao, Shengcheng Han, Yingdian Wang.

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
