## [Decision Letter · Decision Letter 0]

30 Mar 2020

Dear Dr Han,

Thank you very much for submitting your Research Article entitled 'Auxin apical dominance governed by OsAsp1-OsTIF1 complex determines the distinctive development of rice caryopses on different branches' to PLOS Genetics. Your manuscript was fully evaluated at the editorial level and by independent peer reviewers. The reviewers appreciated the attention to an important problem, but raised some substantial concerns about the current manuscript. Based on the reviews, we will not be able to accept this version of the manuscript, but we would be willing to review again a much-revised version. We cannot, of course, promise publication at that time.

If you decide to revise the manuscript for further consideration at PLOS Genetics, please aim to resubmit within the next 60 days, unless it will take extra time to address the concerns of the reviewers, in which case we would appreciate an expected resubmission date by email to plosgenetics@plos.org.

[LINK]

We are sorry that we cannot be more positive about your manuscript at this stage. Please do not hesitate to contact us if you have any concerns or questions.

Yours sincerely,

Lucia Strader, PhD

Guest Editor

PLOS Genetics

Gregory P. Copenhaver

Editor-in-Chief

PLOS Genetics

Reviewer's Responses to Questions

**Comments to the Authors:**

Reviewer #1: This manuscript by Chang et al describes OsAsp1 negatively regulates OsTIF1 through different auxin levels to determine the rice caryopses development in different branches. More interesting results show that, IAA could induce OsTIF1 moving from ER to nucleus to inhibit the interaction formation of OsAsp1-OsTIF1, which as a feedback mechanism to regulate the auxin levels during caryopses development. The experimental design and work are overall well executed, and the mechanism of how caryopses development in different branches are well elucidated. However, there are still some questions and problems need to be supported and addressed.

1- P243-244: “The authors back-crossing the T-DNA insertion lines with WT to conclude both asp1-1 and asp1-2 are single-insertion and homozygous-lethal mutants”. The author should provide the Chi-square statistic test results to support the mutant is a recessive mutation at a single nuclear locus.

2- P244-246: “homozygous-lethal mutants, because only about two-third of spikelets developed into fertilized seeds and no homozygous were found in these seeds (Figure 1E and 1F)”. There are two questions. Firstly, two-thirds of spikelets developed into seeds does not match with the Fig1 F, which shows 75% and 72% are fertilized ovary, only one-fourth unfertilized ovary. Secondly, the author should plant more F2 seedlings to do genotyping, to confirm there is no homozygous, other than just calculate the fertilized seeds. It is hard to understand how the author conclude they are homozygous-lethal mutants.

3- P247-248, “that plant height and spikelet number per panicle were just slightly lower in both asp1-1+/– and asp1-2+/– than in WT”. There is significant difference from the statics results why the author conclude slightly lower. The author should use qRT-PCR to analysis the asp1-1+/– and asp1-2+/– RNA expression levels comparing with the WT, which could possible explain why the two T-DNA insertion lines showing different phenotypes in the following results.

4- P290-291, “However, in asp1-1+/–, the expression peaks of OsTAA1 in both CPB and CSB occurred at 3 DAF, which is lower and earlier than in WT”. This conclusion don’t match with the Fig 3C, which show that expression peaks are higher in asp1-1+/– at 3 DAF.

5- P292-294, “the expression pattern of OsYUCCA1 is similar between WT and asp1-1+/– in both CPB and CSB at 5 and 10 DAF, except that OsYUCCA1 expression is lower in CSB of asp1-1+/– as compared to WT (Supplementary Figure 6)”. This conclusion can not match with the figure, because there is no significant difference in CPB of WT and asp1-1+/– at 5 DAF.

6- Fig5D, there are some problems about the Co-IP experiment, causing there is not showing which antibody used as IP from the figures or methods , and there are no anti-HA protein bands. The author should use anti-GFP to do IP, then using anti-HA to detect if there are bands with or without IAA.

7- Since the manuscript conclude the OsAsp1 as a potential target for yield improvement in rice. It is easy to check the yield per plant of asp1-1+/– and asp1-2+/– comparing with the WT.

8- There are over-expression liens of OsAsp1 in the manuscript used in Fig3. Do the authors check the caryopsis phenotypes in the OE-OsAsp1 lines? These results should be more convince for the conclusion from the manuscript.

Reviewer #2: Chang et al. aims to decipher the mechanistic basis of developmental differences between caryopses located on apical primary branches (CPB) and proximal secondary branches (CSB) in the present manuscript, “Auxin apical dominance governed by OsAsp1-OsTIF1 complex determines the distinctive development of rice caryopses on different branches”. This is very relevant in the context increasing rice yield and productivity, as caryopses on secondary branches have smaller grains with poor grain filling. The authors identified OsAsp1 as a potential candidate gene mediating the observed differences between caryopses on CPB and CSB through transcriptomic comparisons and genetic analyses. They found important role of auxin in the observed differences, and suggested that OsAsp1 induces level of auxin in CPB after heading by binding transcriptional inhibitor of auxin biosynthesis OsTIF1, and thus increasing the levels of OsTAA1. They further reported auxin mediated feedback regulation of the process through translocation of OsTIF1 from ER to nucleus. They also reported the role of MADS29 in regulating the transcription of OsAsp1 by auxin. This could be a very useful study to optimize caryopsis development of secondary branches for yield increases. However, there are a few logical questions regarding the proposed mechanism that need to be addressed.

Authors suggested that high levels of auxin in CPB i.e. auxin apical dominance is key to better grain filling and grain size on CPB compared to CSB. asp mutants showed remarkably low levels or no auxin in both CPB and CSB at different days. The mutant still showed similar (in CPB) or better (in CSB) embryo and endosperm development. How can this be explained if auxin is the key to the studied process?

Further, asp mutant showed phenotypic effects only on CSB, and no effects on CPB (for grain weight, ovary development and size, endosperm cellularization, and endosperm development). This needs to be explained and justified.

What could be the mechanism to achieve initial high levels of IAA that induces OsAsp1 via MADS29? Doesn’t the initial high levels of IAA to induce OsAsp1 require binding of TIF by Asp1? Can it be explained by temporal regulation, or pollination induced IAA level in not dependent upon MADS20, OsAsp1, OsTIF1 and OsTAA1? The initial high levels of IAA, IAA-induced levels of TAA1 to have high IAA levels, and IAA-mediated feedback regulation of the model could possibly be explained more systematically, possibly taking into account temporal and spatial regulation.

What could be the reason of very high expression levels of CycB1 in CSB at 3DAF (Figure 2D)? And even at 5DAF, CSB had comparable expression of CycB1 to CPB? The expression levels of TAA1 in WT and mutant is also not fitting well with IAA content (Figure 3 A,B).

The mechanism of Asp1 and TIF function doesn't explain the increased expression levels of TIF1 and TIF2 in asp1 mutant background (Figure 4B).

The usage of two different auxins IAA and 2,4 D for different experiments, and different rice genotypes for different experiments shall be justified.

**Have all data underlying the figures and results presented in the manuscript been provided?**

Reviewer #1: Yes

Reviewer #2: Yes

PLOS authors have the option to publish the peer review history of their article (what does this mean?). If published, this will include your full peer review and any attached files.

Reviewer #1: No

Reviewer #2: No

---

## [Decision Letter · Decision Letter 1]

23 Jun 2020

Dear Dr Han,

Thank you very much for submitting your Research Article entitled 'Auxin apical dominance governed by OsAsp1-OsTIF1 complex determines the distinctive development of rice caryopses on different branches' to PLOS Genetics. Your manuscript was fully evaluated at the editorial level and by independent peer reviewers. The reviewers appreciated the attention to an important topic but identified some aspects of the manuscript that should be improved.

We therefore ask you to modify the manuscript according to the review recommendations before we can consider your manuscript for acceptance. Your revisions should address the specific points made by each reviewer.

[LINK]

Yours sincerely,

Lucia Strader, PhD

Guest Editor

PLOS Genetics

Gregory P. Copenhaver

Editor-in-Chief

PLOS Genetics

Reviewer's Responses to Questions

**Comments to the Authors:**

Reviewer #1: The authors have answered the most questions, did some experiments to update the figures, and also changed the related expression and explain in the manuscript. However, there are still some questions need to be explained.

1- The are two T-DNA insertion lines used in the manuscript, which display different phenotypes in the Sup Fig3. The author did not detect the RNA levels in asp1-1+/– and asp1-2+/ , and the explain is that “he RNA level in asp1-2+/- is not detected because the following experiments are focused on asp1-1+/- in this study.” Why you focused on asp1-1+/- in this study? Since you did not know the RNA levels in the asp1-1+/ and asp1-2+/ comparing with WT.

2- P186-189, Why the OsTAA1 expression peaks occurred at 3 DAF in asp1-1+/- but at 5 DAF in WT? And why higher peak in CPB at WT, but inverse in asp1-1+/- which display higher in CSB? The author should give some explain rather than just rephrased the sentence in the manuscript.

3- P189-192, The author should give a little bit explain why the OsYUCCA1 expression pattern display different phenotype in the WT and asp1-1+/-, not just reorganize the sentence.

Reviewer #2: In the revised version of the manuscript, authors have tried to address the reviewers’ comments for the original version. However, I think experimental evidence and/or logical explanation supported with available literature are needed, at least for some key aspects of the manuscript.

In response to my comment regarding asp mutants showing similar (in CPB) or better (in CSB) embryo and endosperm development despite remarkably low levels or no auxin in both CPB and CSB, authors suggest that low level of auxin is sufficient to trigger embryo and endosperm development. Since this statement is a key to the basic concept of the manuscript, experimental justification of the statement or logical explanation well supported with available literature shall be provided. I still wonder if that very low levels of IAA in asp1 mutant can mediate similar or better cell proliferation than control having very high auxin levels.

Further, authors also argue that it is the difference in the auxin levels, and not the absolute concentration, is critical for maintaining the development difference of CPB and CSB. There should be some more gravity and support to the statement. Is there any threshold to the relative difference in the auxin levels or any minor difference would also work? This is particularly important for asp mutant which has extremely low auxin levels. Further, exogenous auxin treatment of CSB, leading to disappearance of grain weight differences between CPB and CSB, shall be confirmed for the disappearance of the difference in the auxin levels.

In response to my comment regarding asp mutant showed phenotypic effects only on CSB, and no effects on CPB, the authors explain the grain weight differences in CPB and CSB of mutants. However, the question was more towards figure 2 A and B, where wild type and mutant CPB appear very similar for ovary development and size, endosperm cellularization, and endosperm development. This needs to be explained that why asp mutants effect ovary and endosperm development specifically on CSB, and not on CPB. Figure 1G also shows that no significant difference between CPB of wild type and asp1-1. There should be some explanation of why the phenotypic effects is specific to CSB, and not CPB.

Considering the complexity of the process, I think some logical discussion about possibility of temporal and spatial regulation between CPB and CSB may help the manuscript.

Regarding my comment on discrepancy in expression levels CycB1, I understand that the expression of cell-cycle genes is not the key point of the study. However, the expression peak of cell-cycle genes in CPB shall overlap with auxin peak and Asp1 expression peak in CPB to have more confidence in the model, as auxin peak has been suggested to be a key to differential embryo and endosperm development via cell proliferation.

I still wonder for a reason to use two different auxins: 2,4D for in vitro culture of spikelets, and IAA for treatment of spikelets on panicle.

The manuscript shall also be thoroughly checked for language and sentence structure.

**Have all data underlying the figures and results presented in the manuscript been provided?**

Reviewer #1: Yes

Reviewer #2: None

PLOS authors have the option to publish the peer review history of their article (what does this mean?). If published, this will include your full peer review and any attached files.

Reviewer #1: No

Reviewer #2: No

---

## [Decision Letter · Decision Letter 2]

26 Sep 2020

Dear Dr Han,

We are pleased to inform you that your manuscript entitled "Auxin apical dominance governed by the OsAsp1-OsTIF1 complex determines distinctive rice caryopses development on different branches" has been editorially accepted for publication in PLOS Genetics. Congratulations!

Yours sincerely,

Lucia Strader, PhD

Guest Editor

PLOS Genetics

Gregory P. Copenhaver

Editor-in-Chief

PLOS Genetics

Comments from the reviewers (if applicable):

Reviewer's Responses to Questions

**Comments to the Authors:**

Reviewer #1: The author has answered the most question. No any further concerns.

Reviewer #2: The authors seem to have reasonably addressed my comments for the previous version of the manuscript, though a tracked change file would have been of help to see the changes in the manuscript. Nonetheless, the current version is much improved and streamlined than the initial submission.

**Have all data underlying the figures and results presented in the manuscript been provided?**

Reviewer #1: Yes

Reviewer #2: None

PLOS authors have the option to publish the peer review history of their article (what does this mean?). If published, this will include your full peer review and any attached files.

Reviewer #1: No

Reviewer #2: No

**Data Deposition**

http://datadryad.org/submit?journalID=pgenetics&manu=PGENETICS-D-20-00169R2

**Press Queries**

---

## [Editor Report · Acceptance letter]

15 Oct 2020

PGENETICS-D-20-00169R2 

Auxin apical dominance governed by the OsAsp1-OsTIF1 complex determines distinctive rice caryopses development on different branches 

Dear Dr Han, 

We are pleased to inform you that your manuscript entitled "Auxin apical dominance governed by the OsAsp1-OsTIF1 complex determines distinctive rice caryopses development on different branches" has been formally accepted for publication in PLOS Genetics! Your manuscript is now with our production department and you will be notified of the publication date in due course.

With kind regards,

Matt Lyles

PLOS Genetics

On behalf of:
